# Knowledge of and Attitude towards Epilepsy among the Jordanian Community

**DOI:** 10.3390/healthcare10081567

**Published:** 2022-08-18

**Authors:** Sawsan M. A. Abuhamdah, Abdallah Y. Naser, Mohammed Ahmed R. Abualshaar

**Affiliations:** 1Department of Pharmaceutical Sciences, College of Pharmacy, Al-Ain University, Abu Dhabi Campus, Abu Dhabi P.O. Box 112612, United Arab Emirates; 2Department of Biopharmaceutics and Clinical Pharmacy, Faculty of Pharmacy, The University of Jordan, Amman 11942, Jordan; 3Department of Applied Pharmaceutical Sciences and Clinical Pharmacy, Faculty of Pharmacy, Isra University, Amman 11622, Jordan; 4Quality Assurance Department, Hikma Pharmaceuticals, Amman 11118, Jordan

**Keywords:** attitude, community, epilepsy, Jordan, knowledge

## Abstract

Background: Epilepsy is a disorder characterized by recurring seizures that do not have an immediate identifiable cause. It is a disorder with complex symptoms and a wide range of risk factors, with age, genetics, and origin being the most prevalent variations. This study aimed to evaluate the knowledge of and attitude towards epilepsy among the Jordanian community. Method: An online cross-sectional study using a self-administered questionnaire was conducted between 29 March and 15 May 2022 in Jordan. In this study, three previously validated questionnaire items were adapted and employed. Binary logistic regression was applied to identify predictors of good knowledge and a positive attitude. Results: A total of 689 participants were involved in this study. A weak level of knowledge about epilepsy was observed among the study participants (35.3%). The participants showed a moderately positive attitude towards epilepsy (63.3%). Being female, holding a bachelor’s degree, knowing anyone who had epilepsy and seeing anyone having an epileptic seizure were factors that positively affected participants’ knowledge about epilepsy. Being aged between 24 and 29 years or being divorced were factors that affected the participants’ attitudes negatively towards epilepsy. Conclusion: The study’s participants had limited knowledge of epilepsy and a favorable attitude toward it. The community’s understanding of epilepsy and attitude toward epilepsy patients should be improved by an informed educational effort on the part of various media platforms. All facets of the community, including parents, should be the focus of these initiatives.

## 1. Introduction

Epilepsy is a brain disorder characterized by recurring (more than one) seizures that do not have an immediate identifiable cause. It is a disorder with complex symptoms and a wide range of risk factors, with age, genetics, and origin being the most prevalent variations [1]. As was previously stated, epilepsy is one of the most prevalent neurological disorders, with a prevalence of 5–8 per 1000 persons and a lifetime risk of 3.0% [2]. Research found that in undeveloped countries, incidence rates were approximately 190 per 100,000 persons annually [3]. Other studies showed that the incidence of epilepsy is highest in the first year of life and gradually declines until the age of 70, at which point it sharply increases [4].

Approximately 30% of patients continue to experience seizures despite receiving therapy, which may cause social discomfort and discrimination with the misunderstanding and resulting social stigma [5]. People who have uncontrolled seizures are constantly at risk of becoming unconscious, as well as falling and getting hurt in public, and as a result, the need to learn about and raise awareness of epilepsy has increased [6].

Studies on knowledge and attitudes about epilepsy have been conducted in both developed and western countries, as well as the Arab world. In Saudi Arabia, for instance, a study involving 1044 participants found that the vast majority of participants had heard of epilepsy, despite the fact that 60.7% of them demonstrated ignorance of the condition and an inappropriate attitude when interacting with someone having a seizure [7]. Another study conducted in the United Arab Emirates (UAE) in 1998 found that the majority had heard of epilepsy. It also revealed that education level and age have a positive impact on knowledge of the condition. Different people hold different views regarding treatment options, with some believing there is no cure and others supporting cautery and faith healing [8]. The majority of participants in a survey study conducted in the UAE in 2016 had heard of epilepsy, and 80 % had a positive attitude toward societal acceptance. Despite that, 14 % believed that epilepsy patients should be kept to themselves [9]. All of the participants in a study that was conducted in Egypt had heard about epilepsy. Families with an epileptic patient had better attitudes, according to a specific comparison, than families without one. Additionally, it demonstrated that Egyptian students had higher levels of social acceptance and positive attitudes [10].

A previous study in Kuwait found that most individuals were aware of epilepsy and that around half of them had interacted with someone who had it. A greater number disapproved of marrying people with epilepsy, while a smaller percentage believed that placing something in the patient’s mouth to stop harm was ignorant [11]. Despite the fact that more than 75% of participants in a previous survey in Jordan in 2005 rejected the idea of getting married to someone who had epilepsy, the majority of participants had favorable attitudes towards epilepsy patients. Therefore, it was found that Jordanians had less knowledge and attitudes towards epilepsy than individuals in other Arab countries, but more than people in western countries [12]. Epilepsy is a disorder about which very little is known, and cultural views regarding the condition might differ significantly [13,14,15,16]. Despite that epilepsy knowledge, attitudes, and beliefs have improved significantly, there are still misconceptions about the condition [13,17]. In many communities throughout history, false beliefs regarding epilepsy have resulted in stigmatizing attitudes against those who have it [18]. Therefore, it is critical to evaluate public knowledge of the condition and ascertain whether Jordanians are aware of how to react when someone is having a seizure.

There have only been a few studies conducted in the past that looked at public knowledge in Jordan regarding epilepsy and its attitudes. Prior research has focused on specific populations, such as teachers, epilepsy patients’ parents, epilepsy patients themselves, or schoolchildren [19,20,21,22]. The purpose of this study was to evaluate the knowledge of and attitude towards epilepsy among the Jordanian community.

## 2. Method

### 2.1. Study Design

An online cross-sectional study using a self-administered questionnaire was conducted between 29 March and 15 May 2022 in Jordan.

### 2.2. Sampling Strategy

We used the convenience sample technique to recruit study participants. Jordanians who were currently living in Jordan and were willing to participate in the study made up the study population. To be considered, participants had to reside in Jordan and be at least 18 years old. Those who fit these criteria were invited to participate in the study. These inclusion criteria were mentioned in the cover letter of the survey. Through social media platforms such as WhatsApp, Twitter, and Facebook, we invited the study’s target study population to participate. The study’s aims and objectives were clearly mentioned at the start of the survey. They were encouraged to participate in the survey that was designed to evaluate public awareness and attitudes toward epilepsy.

### 2.3. Questionnaire Tool

In this study, three previously validated questionnaire items were adapted and employed. Young et al. and Diby et al. used two previously developed questionnaires to examine public knowledge about epilepsy [23,24]. Students at Fanshawe College in London, Ontario, Canada, were given the questionnaire instrument by Young et al. We used eight questions from Young et al.’s questionnaire instrument. The first four items asked the participants if they had ever heard or read about epilepsy, if they had more than one seizure after the age of five, if they knew or had ever known somebody with epilepsy, and if they had ever witnessed someone having an epileptic seizure. The remaining four items were multiple choice questions (MCQs) that tested participants’ epilepsy knowledge (epilepsy incidence rate, causes of epilepsy, characteristics of epileptic attacks, and their thoughts about drug therapy for epilepsy). We added one knowledge question adapted from a study by Diby et al. to the four items adapted from Young et al. to explore participants’ knowledge. This item tested the participants’ knowledge of epileptic seizure signs and symptoms. The number of correct answers provided by participants was used to assess their knowledge. We had 15 correct answers based on our customized knowledge assessment scale. Each correct answer was given a one-point weighting, with a maximum score of 15. The higher the score, the better the participant’s knowledge of epilepsy.

The second section of our survey measured the participants’ attitudes toward epilepsy. Our attitude assessment scale was adapted from Wubetu et al.’s research [25]. They have previously checked the completeness, accuracy, clarity, and consistency of their questionnaire tool. There were 12 items on the attitude scale (4-point Likert scale format). Participants’ responses ranged from strongly agree, “given a weight of four” to strongly disagree, “given a weight of one”. According to this scoring scale, the highest possible score is 48. Items seven and eleven were negatively worded. Therefore, they were negatively scored upon the score analysis. Cronbach’s alpha was used to test the attitude assessment scale’s reliability. The assessment scale’s Cronbach’s alpha value was 0.657, indicating acceptable internal consistency. The third section of the questionnaire focused on the demographic characteristics of the participants (age, gender, education level, marital status, employment status, and income category).

The study questionnaire was translated into Arabic using the forward-backward translation technique. The forward translation was mainly focused on conceptual translation. The backward translation followed the forward translation. The original questionnaires and the back-translated draft of the questionnaire were compared. The questions were then confirmed to be simple to understand and to reflect the study’s objectives in a pilot study that involved a small number of participants from the general public. They were asked to complete the questionnaire tool in person and were also asked if any of the questions were considered unacceptable, offensive, or difficult to understand.

### 2.4. Sample Size

Survey Monkey website was used to estimate the required sample size for this study [26]. The required sample size was 385 participants, based on a confidence interval of 95%, a standard deviation of 0.5, a margin of error of 5%, and population on 10,414,179 persons.

### 2.5. Statistical Analysis

Data were analyzed using Statistical Package for Social Science (SPSS) software, version 27 (IBM Corp, Armonk, NY, USA). The mean and standard deviation (SD) of continuous data were presented. Frequencies and percentages were used to report categorical variables. The researchers employed binary logistic regression analysis to look for indicators of good epilepsy knowledge and a positive attitude. For the binary logistic regression, good epilepsy knowledge was defined as a total score above the study participants’ mean knowledge score (5.3; SD: 2.6). A total score above the mean attitude score of the study participants (30.4; SD: 4.7) was defined as having a positive attitude toward epilepsy. A confidence interval of 95% (*p* < 0.05) was applied to represent the statistical significance of the results, and the level of significance was predetermined as 5%.

## 3. Results

### 3.1. Participants’ Characteristics

A total of 689 participants were involved in this study. More than half of them (59.1%) were females. Around one-third of the study participants (30.2%) were aged 19–23 years. A total of 39.3% of them were married and more than half of them (53.8%) reported that they held bachelor’s degrees. Around one-third of the study participants (28.4%) reported that they are working outside the healthcare sector. A total of 38.9% of them reported that their monthly income category was less than 500 JD.

More than half of the study participants (59.5%) reported hearing or reading about epilepsy. Around one-quarter of the study participants (24.2%) reported that have had more than one seizure after the age of 5 years. A total of 42.8% of the study participants reported that they knew someone who had epilepsy. More than half of the study participants (53.4%) confirmed that they had seen someone having an epileptic seizure in the past, Table 1.

### 3.2. Knowledge about Epilepsy

Overall, the study participants showed a weak level of knowledge about epilepsy. The proportion of participants who answered the questions correctly ranged from 15.2% to 53.0%. Less than one-quarter of the study participants knew the correct incidence rate of epilepsy. The proportion of participants who were able to identify the right causes of epilepsy ranged from 15.2% (for stroke) to 40.9% (for inherited disease). A proportion of 41.1% (as a period of memory disturbance) to 53.0% (as an episode of behavioral change) of the study participants were able to identify what an epileptic attack looked like. Concerning the participants’ knowledge about drug therapy for epilepsy, only 28.9% of them reported that drug therapy for epilepsy occasionally produces malformations in babies of mothers with epilepsy, and a similar percentage (30.9%) reported that it has advanced significantly over the past 10 years. The proportion of participants who were able to identify the signs and symptoms of an epileptic seizure ranged from 34.7% (for falling down) to 50.2% (for rolling of the eyes), Table 2.

### 3.3. Attitude towards Epilepsy

Our study participants showed a different degree of positive attitude towards patients with epilepsy. The level of agreement concerning having a positive attitude towards patients with epilepsy ranged from 40.5% (for agreeing to live together with epileptics) to 72.6% (for agreeing to shake hands with epileptics), Table 3.

### 3.4. Predictors of Good Knowledge and a Positive Attitude

The study participants showed a weak level of knowledge about epilepsy with a mean score of 5.3 (SD: 2.6) out of a maximum obtainable score of 15 (equal to 35.3% out of a maximum obtainable score). In contrast, they showed a moderately positive attitude towards epilepsy with a mean score of 30.4 (SD: 4.7) out of a maximum obtainable score of 48 (equal to 63.3% out of a maximum obtainable score).

We used binary logistic regression analysis to identify factors that affect participants’ knowledge and attitude towards epilepsy positively, which revealed that being female, holding a bachelor’s degree, knowing anyone who had epilepsy, and having seen someone having an epileptic seizure were factors that positively affected participants’ knowledge about epilepsy (*p* ≤ 0.05). In contrast, being aged between 24 and 29 years or being divorced were factors that affected the participants’ attitude negatively towards epilepsy (*p* ≤ 0.05), Table 4.

## 4. Discussion

This study aimed to understand the knowledge and attitudes of the Jordanian population about epilepsy. The study participants showed a weak level of knowledge about epilepsy, a moderately positive attitude towards epilepsy, and disturbances of awareness depending on factors such as gender, education, and relations with an epileptic patient towards epilepsy. The key study findings of this study are: (1) A weak level of knowledge about epilepsy was observed among the study participants (35.3%), (2) The participants showed a moderately positive attitude towards epilepsy (63.3%), (3) Being female, holding a bachelor’s degree, knowing someone who had epilepsy, and seeing someone having an epileptic seizure were factors that positively affected participants’ knowledge about epilepsy, and (4) Being aged between 24 and 29 years or being divorced were factors that negatively affected the participants’ attitude towards epilepsy.

One of the most prevalent neurological disorders, epilepsy affects a large number of patients globally and has a negative impact on their quality of life, especially in young patients [27]. It is estimated that there are more than 50 million epilepsy sufferers worldwide [28]. Our study showed a weak level (35.3%) of knowledge about epilepsy among the study participants. However, this low level of knowledge in our study is consistent with other studies in the United States, where one out of every four knows about epilepsy [25] as well as the low level of knowledge among the Ethiopian population [25]. Similar results were found in a previous study conducted in Canada, where participants had poor knowledge of the prevalence of epilepsy in the general population, hereditary epilepsy and several other etiologies, the recognition of non-convulsive seizures as a type of epilepsy, and the teratogenicity of antiepileptic medications [23]. A previous study in Jordan revealed different findings that more than 80% of the Jordanian population have knowledge about cases and symptoms of epilepsy [14]. A recent study in Saudi Arabia showed consistent findings that the participants showed a weak level of knowledge about autism (34.7%), with a middle income category being less likely to be knowledgeable about autism compared to others [29].

Despite the fact that education level was a consistent feature in all of these studies, people with higher levels of education had more epilepsy knowledge [30]. The amount of exposure to the idea and concept of epilepsy may vary, as seen by the fact that 53.8 % of our study participants reported having a bachelor’s degree. This suggests that more exposure to the word epilepsy will result in more searches and understanding of it [25]. Similar to a study in the United Kingdom, where negative attitudes toward epilepsy were present but in relatively low numbers [31], our study population (63.3%) had a positive attitude toward the condition. In our study, the level of agreement concerning having a positive attitude towards patients with epilepsy ranged between 40.5% and 72.6%. This was lower than that for a previous study that was conducted in Canada, where agreement levels ranged between 84.0% and 95% [23]. In our study, the lowest level of agreement was for living together with epileptic patients, their employment, and marrying them. Previous studies in Canada and Hungary showed similar findings; the lowest agreement was related to employing epileptic patients [14,23]. The main drivers behind this low agreement level towards the employment of epileptic patients were their safety and abilities. Given that some jobs are not recommended for people with epilepsy, participants who refused to allow an epileptic the chance to work may have been motivated to protect the person with epilepsy [32].

A previous systematic review examined the employment rate among 95 studies, which ranged between 14% and 89%, with a mean adjusted employment rate of 58% [33]. Employability of epileptic patients was significantly influenced by an individual’s ability to work (education level), clinical condition (epilepsy type), and psychological factors (self-determined motivation and family overprotection) [34]. According to earlier research [35], people with epilepsy typically experience problems in marriage, such as decreased chances for marriage, poor marital outcomes, and a decreased quality of married life. This information is relevant to our study’s findings that people without epilepsy were less likely to agree to marry them. This confirms the findings of a previous study in Jordan, where participants in this study showed a less favorable attitude toward marriage to someone with epilepsy [36]. Traditional habits and strong cultural norms and attitudes that Jordanians have regarding marriage may be the cause of this unfavorable attitude [36,37].

Knowledge levels also have an impact on how people behave and feel about those who have epilepsy [38,39]. Contrarily, our survey revealed low levels of awareness regarding epilepsy but a disproportionately favorable attitude toward the condition. Despite population variations, negative attitudes were noted in Vietnam (33%) and Ethiopia (35.6%), respectively [25,40]. These changes in values are thought to be the result of cultural differences that have an impact on how people view those who have epilepsy [41].

In our study, we examined the variables that influence the study participants’ knowledge of epilepsy. Factors that favorably impacted participants’ knowledge of epilepsy included being a woman, having a bachelor’s degree, knowing someone who had epilepsy, and having witnessed someone having an epileptic seizure. Education has a significant impact on the information that influences how epileptic patients are viewed [42], and it is generally accepted that university graduates have more knowledge than those with less education [38]. In contrast, women typically know more about epilepsy than men do [42,43]. In this study, 59.1% of participants were female, making up more than half of the study population. In two further studies [42,44], knowing someone who had epilepsy or having witnessed an epileptic seizure was found to be a contributing factor to behavior and attitudes toward epileptic patients. However, unfavorable traits, including being between the ages of 24 and 29 or being divorced, had a detrimental impact on the participants’ attitudes about epilepsy. Even religion may have an impact on how the general public views those who have epilepsy [45].

Low-educated, elderly, and people with epilepsy exhibit a lack of information and a lack of understanding of their condition [46], which affects their treatment path and how they describe their condition to others, which may have an impact on their mental and social wellbeing. There is a stigma against this group of epileptics, and in order to combat this stigma, it is necessary to learn as much as possible about the causes and motivations of seizure episodes [47]. Children who are exposed to such a circumstance are more likely to experience psychological problems and social problems [48]. If the parents are sufficiently informed about their child’s condition, this largely depends on how they feel about them. This will likely have a good impact on the health of the children, in contrast to ignorance, which would render them defenseless in the face of seizure situations and have a negative impact on the child even as they grow older and are unable to control or recognize their condition [49]. Generally speaking, all facets of society play a significant part in combating this disorder, whether through a private contribution with a family member or a well-known epileptic or a public contribution within the community to ensure increasing the effectiveness of dealing with and understanding people with epilepsy [50].

Two studies in Jordan identified the presence of a negative attitude toward epilepsy [12,36]. These findings highlight a significant issue that needs to be addressed because, in the absence of a solution, this attitude will only worsen people with epilepsy’s conditions and their socioeconomic circumstances, which may affect how well they take their medications [51,52].

The findings of our research showed that there is a knowledge gap among the general public on epilepsy, particularly among males and the elderly. In addition, younger populations were more prone to having a negative attitude towards epileptic patients. Family and friends, the news and entertainment media, the internet, and social media are just a few of the places where people can learn more about epilepsy. These sources’ reliability, meanwhile, varies. Since neurological conditions such as epilepsy have a considerable impact on the productivity of communities, public health experts should pay them more attention. Furthermore, despite being difficult, training medical personnel to deliver better epilepsy care would help raise patients’ quality of life and lower their associated expenses.

This puts pressure on national policymakers and healthcare professionals to enhance the general knowledge and lessen misconceptions about these kinds of disorders. The dissemination of brochures and booklets, awareness campaigns, and the usage of social media sites such as Facebook and Twitter are all advised methods for raising awareness about epilepsy and transforming people’s attitudes towards those who suffer from epilepsy. The community’s understanding of epilepsy and attitude toward epileptic patients should be improved by a concerted educational effort on the part of various media platforms. The goal of educational efforts should be to increase public awareness of the causes, symptoms, and treatments of epilepsy. These initiatives should involve people with epilepsy and their families in public awareness campaigns, to coordinate public awareness campaigns and develop shared messaging, and to make sure that all campaigns incorporate thorough formative research, take audience demographics and health literacy into account, as well as involve mechanisms for evaluation and sustainability [53]. Combating stigma is a key objective of such initiatives that educate and raise awareness of epilepsy among the general public. By that time, society would have improved for epileptics, and only by comprehending the conditions and the right ways to address them would their quality of life have improved.

Future research should focus on developing efficient educational interventions that enhance the public’s knowledge of epilepsy. This includes using entertainment media to promote fewer sensationalistic portrayals and more opportunities for the passive learning of factual facts about epilepsy.

The study does have certain limitations. The establishment of causation between research variables is prohibited by the cross-sectional study design. Another limitation of this type of study (self-reported studies) is the tendency of survey respondents to answer questions in a way that will be viewed favorably by others. We employed a quantitative approach with pre-set responses, which may have hindered the collection of respondents’ opinions in order to provide a range of qualitative data that was nonetheless valuable. Additionally, we collected data via an online survey, so we could have missed some of the targeted population. Finally, because we used a convenience sampling technique to distribute the online questionnaire for our study, we were unable to quantify the response rate. Therefore, our findings should be interpreted carefully.

## 5. Conclusions

The study’s participants had limited knowledge of epilepsy and a favorable attitude toward it. The community’s understanding of epilepsy and attitude toward epilepsy patients should be improved by an informed educational effort on the part of various media platforms. All facets of the community, including parents, should be the focus of these initiatives.

## Figures and Tables

**Table 1 healthcare-10-01567-t001:** Characteristics of the study participants.

Variable	Frequency	Percentage
Gender		
Female	407	59.1%
Age category		
18–23 years	208	30.2%
24–29 years	106	15.4%
30–34 years	147	21.3%
35–39 years	92	13.4%
40–45 years	81	11.8%
46 years and over	55	8.0%
Marital status		
Single	241	35.0%
Married	271	39.3%
Divorced	130	18.9%
Widowed	47	6.8%
Education level		
Secondary school or lower	191	27.7%
Bachelor degree	371	53.8%
Higher studies	127	18.4%
Employment status		
Unemployed	107	15.5%
Student	205	29.8%
Retired	116	16.8%
Working in the healthcare sector	65	9.4%
Working outside the healthcare sector	196	28.4%
Monthly income category		
Less than 500 JD *	268	38.9%
500–1000 JD	204	29.6%
1000–1500 JD	168	24.4%
1500 JD and above	49	7.1%
Have you ever heard or read about epilepsy? (yes)	410	59.5%
Have you ever had more than one seizure after the age of 5 years? (yes)	167	24.2%
Do you know or have you ever known anyone who had epilepsy? (yes)	295	42.8%
Have you ever seen anyone having an epileptic seizure? (yes)	368	53.4%

JD: Jordanian Dinar; * one JD equals 1.43 USD.

**Table 2 healthcare-10-01567-t002:** Participants’ responses to knowledge items.

No.	Variable	Frequency	Percentage
1	Epilepsy occurs in …		
	One in every 100 people *	162	23.5%
	One in every 1000 people	196	28.4%
	One in every 10,000 people	161	23.4%
	One in every 50,000 people	109	15.8%
	One in every 1,000,000 people	61	8.9%
2	What do you think causes epilepsy? (more than one answer could be chosen)		
	Accidents *	150	21.8%
	Inherited disease *	282	40.9%
	Insanity or other mental illness	281	40.8%
	Brain tumors *	202	29.3%
	Birth defects *	145	21.0%
	Stroke *	105	15.2%
	Don’t know	65	9.4%
3	What do you think an epileptic attack is? (more than one answer could be chosen)		
	A loss of consciousness *	310	45.0%
	An episode of behavioral change *	365	53.0%
	A period of memory disturbance *	283	41.1%
	Don’t know	76	11.0%
4	What do you think about drug therapy for epilepsy? (more than one answer could be chosen)		
	It is seldom effective in controlling seizures	186	27.0%
	It is best given as two or more drugs that work together	208	30.2%
	It has advanced significantly over the past 10 years *	213	30.9%
	It occasionally produces malformations in babies of mothers with epilepsy *	199	28.9%
	It can be stopped abruptly after seizures are controlled for a year.	120	17.4%
	Don’t know	129	18.7%
5	What are the signs and symptoms of epileptic seizures you know of? (more than one answer could be chosen)		
	Falling down *	239	34.7%
	Rolling of the eyes *	346	50.2%
	Foaming of the mouth *	337	48.9%
	Uncontrolled jerking movement of the arms and legs *	284	41.2%

* The correct answer(s).

**Table 3 healthcare-10-01567-t003:** Participants responses to attitude items.

No.	Variable	Strongly Disagree	Disagree	Agree	Strongly Agree
1	Agree to work with epileptics	23.1%	35.6%	35.0%	6.4%
2	Agree to have close relation with epileptics	14.4%	37.0%	40.3%	8.3%
3	Agree to live together with epileptics	18.7%	40.8%	34.5%	6.0%
4	Epileptics should not be isolated	16.3%	31.9%	37.4%	14.4%
5	Epileptics can manage their family	15.7%	30.3%	44.1%	9.9%
6	Agree to shake hands with epileptics	6.1%	21.3%	55.2%	17.4%
7	Keep child from contacting epileptics	14.4%	44.1%	31.9%	9.6%
8	Agree to recruit epileptics as a servant	13.2%	41.8%	36.3%	8.7%
9	Agree with family member marrying epileptics	19.2%	40.2%	33.7%	7.0%
10	Epilepsy is a treatable disease	8.4%	25.3%	52.1%	14.2%
11	Epileptics should not learn in schools	21.9%	39.5%	26.9%	11.8%
12	Epileptics can lead a healthy lifestyle	7.8%	20.2%	53.1%	18.9%

**Table 4 healthcare-10-01567-t004:** Factors affecting participants’ knowledge and attitude towards epilepsy positively.

Variable	Odds Ratio for Being Knowledgeable about Epilepsy (95% Confidence Interval)	Odds Ratio for Having Positive Attitude towards Epilepsy (95% Confidence Interval)
Gender		
Male (Reference group)	1.00	1.00
Female	1.77 (1.29–2.44) **	1.35 (0.99–1.83)
Age category		
18–23 years (Reference group)	1.00	1.00
24–29 years	1.17 (0.77–1.78)	0.60 (0.39–0.91) *
30–34 years	1.03 (0.71–1.49)	0.82 (0.57–1.18)
35–39 years	1.28 (0.82–2.00)	0.86 (0.55–1.33)
40–45 years	0.89 (0.55–1.43)	0.83 (0.52–1.32)
46 years and over	0.47 (0.25–0.90) *	1.41 (0.80–2.47)
Marital status		
Single (Reference group)	1.00	1.00
Married	1.37 (1.00–1.87)	0.94 (0.69–1.28)
Divorced	0.76 (0.51–1.13)	0.61 (0.42–0.90) *
Widowed	0.67 (0.35–1.27)	1.25 (0.69–2.28)
Education level		
Secondary school or lower (Reference group)	1.00	1.00
Bachelor degree	1.45 (1.06–1.97) *	1.10 (0.82–1.49)
Higher studies	0.61 (0.40–0.92) *	0.75 (0.51–1.11)
Employment status		
Retired (Reference group)	1.00	1.00
Unemployed	0.64 (0.41–1.01)	1.31 (0.87–1.99)
Working outside the healthcare sector	0.94 (0.67–1.32)	0.81 (0.58–1.13)
Student	1.21 (0.87–1.69)	1.36 (0.98–1.89)
Working in the healthcare sector	1.33 (0.80–2.23)	1.00 (0.60–1.67)
Monthly income category		
Less than 500 JD (Reference group)	1.00	1.00
500–1000 JD	1.33 (0.95–1.86)	0.77 (0.55–1.06)
1000–1500 JD	0.86 (0.60–1.24)	0.76 (0.54–1.07)
1500 JD and above	0.85 (0.46–1.56)	1.13 (0.63–2.03)
Have you ever had more than one seizure after the age of 5 years?		
No (Reference group)	1.00	1.00
Yes	0.97 (0.68–1.39)	0.75 (0.53–1.06)
Do you know or have you ever known anyone who had epilepsy?		
No (Reference group)	1.00	1.00
Yes	1.38 (1.01–1.89) *	1.10 (0.81–1.48)
Have you ever seen anyone having an epileptic seizure?		
No (Reference group)	1.00	1.00
Yes	1.49 (1.09–2.03) *	0.88 (0.66–1.19)

* *p* ≤ 0.05; ** *p* ≤ 0.001.

## Data Availability

Not applicable.

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
