# Peer review of "Knowledge of and Attitude towards Epilepsy among the Jordanian Community"

_healthcare, 2022, doi:10.3390/healthcare10081567_

Round 1
Reviewer 1 Report
The Manuscript investigates the Knowledge of and attitude towards epilepsy among the Jordanian community through survey. Please find the following comments regarding the manuscript;
1- Title: is well written
2- Abstract: There is no need to describe in details the sources of questionnaire.
3- Introduction; L44: developing countries
- The introduction does not justify well the study aim, or mention the limitations in the current evidence.
4- Methods: what is the language of the used survey?
- Please cite the info of calculating required power sample size.
5- Discussion: The discussion is poor. Why the Jordanians have poor knowledge. How can we use the results of your study. There is no recommendations for society, steak holders, policy makers, or research.
Author Response
Response to reviewers:
Manuscript ID; healthcare-1820633
Title: Knowledge of and attitude towards epilepsy among the Jordanian community
Corresponding Author: Dr. Sawsan Abu Hamdah
Dear Editor,
Thank you for the opportunity to revise and resubmit our manuscript based on the reviewers’ comments. Please find below our itemized point-by-point responses to the journal requirements and reviewers’ comments.
Reviewer 1:
The Manuscript investigates the Knowledge of and attitude towards epilepsy among the Jordanian community through survey. Please find the following comments regarding the manuscript;
- First of all, we would like to thank the reviewer for his time and efforts in reviewing our manuscript.
1- Title: is well written
2- Abstract: There is no need to describe in details the sources of questionnaire.
- Thank you for this comment. We have now removed the details the sources of questionnaire based on the reviewer comment, see page 1, lines 21-24.
3- Introduction; L44: developing countries
- The introduction does not justify well the study aim, or mention the limitations in the current evidence.
- Thank you for this comment. We have now addressed the reviewer comment and added justification for the current study and limitations of previous study, see pages 2-3, lines 97-107.
4- Methods: what is the language of the used survey?
- Thank you for this comment. The language used for this survey study was Arabic, which is the official language of the country. We have now clarified this in the method section, see page 4, lines 162-169.
- Please cite the info of calculating required power sample size.
- Thank you for this comment. We have now cited the info of calculating required power sample size, see page 4, line 171.
5- Discussion: The discussion is poor. Why the Jordanians have poor knowledge. How can we use the results of your study. There is no recommendations for society, steak holders, policy makers, or research.
- Thank you for this comment. Based on the reviewer comment, we have now enriched our discussion with more comparison to previous studies and interpretation (see pages 9-10, lines 276-303) and provided further recommendations for policymakers, healthcare professionals, government, and future research, see page11, lines 344- 376.
Reviewer 2 Report
Major issues
#1. It is hard to for me to read some parts due to ungrammatical phrases. Please use an professional English proofreading service. For example, repeated use of “On the other hand” without using “ On the one hand”.
Minor issues
#1. Please number each question. For example, 1) if they had ever heard or read about epilepsy; 2) if…
Author Response
Response to reviewers:
Manuscript ID; healthcare-1820633
Title: Knowledge of and attitude towards epilepsy among the Jordanian community
Corresponding Author: Dr. Sawsan Abu Hamdah
Dear Editor,
Thank you for the opportunity to revise and resubmit our manuscript based on the reviewers’ comments. Please find below our itemized point-by-point responses to the journal requirements and reviewers’ comments.
Reviewer 2:
#1. It is hard to for me to read some parts due to ungrammatical phrases. Please use an professional English proofreading service. For example, repeated use of “On the other hand” without using “ On the one hand”.
- First of all, we would like to thank the reviewer for his time and efforts in reviewing our manuscript. Based on the reviewer comment, we have now conducted full proofreading for our study.
Minor issues
#1. Please number each question. For example, 1) if they had ever heard or read about epilepsy; 2) if…
- Thank you for this comment. We have now number the questions based on the reviewer comment.
Reviewer 3 Report
Sawsan MA Abuhamdah et al. studied knowledge of and attitude towards epilepsy among the Jordanian community. The study is important from the public health care view and advocacy for people with epilepsy. However, most probably won't attract the broader public.
Authors should improve their manuscript. In general, they should write more concise. Sentences are very long, sometimes the whole paragraph. Authors often repeat themself in the text. The manuscript is hard to read in its present form.
Major:
Methods:
1. there is no information on how many invitations they send and what was the response rate
2. how did they calculate the sample size. What was their reference?
3. questionnaires should be described in the introduction
Discussion:
1. no comparison with the previous two studies - what is new, what is known and what differs.
2. Authors didn't critically evaluate their work. All limitations are described in one line.
3. Some claims in the text are not suitable for a scientific manuscript.
E.g. line 265-266 "All of these issues can be resolved simply by raising awareness of epilepsy and including those who have it in educational sessions." Unfortunately, it is not as simple as the authors claim.
Line 271-273, they call for action from the government. However, there is no data, and the authors did not provide any reference to support this.
The manuscript in the present form is not suitable for publication.
Author Response
Response to reviewers:
Manuscript ID; healthcare-1820633
Title: Knowledge of and attitude towards epilepsy among the Jordanian community
Corresponding Author: Dr. Sawsan Abu Hamdah
Dear Editor,
Thank you for the opportunity to revise and resubmit our manuscript based on the reviewers’ comments. Please find below our itemized point-by-point responses to the journal requirements and reviewers’ comments.
Reviewer 3:
Sawsan MA Abuhamdah et al. studied knowledge of and attitude towards epilepsy among the Jordanian community. The study is important from the public health care view and advocacy for people with epilepsy. However, most probably won't attract the broader public.
Authors should improve their manuscript. In general, they should write more concise. Sentences are very long, sometimes the whole paragraph. Authors often repeat themself in the text. The manuscript is hard to read in its present form.
- First of all, we would like to thank the reviewer for his time and efforts in reviewing our manuscript. We have now checked our manuscript to address the reviewer comment and rephrase any inappropriate paragraphs.
Major:
Methods:
- there is no information on how many invitations they send and what was the response rate
- Thank you for this comment. Unfortunately, we were not able to estimate the response rate in our study as we distributed our online study questionnaire link using social media website. We have now highlighted this point in the limitations section, see page 12, lines 395-397.
- how did they calculate the sample size. What was their reference?
- Thank you for this comment. We have now cited the website (which is Survey Monkey) that we used to estimate our sample size, see page 4, line 171. We used the number of the Jordanian population (which is 10,414,000 persons) as the reference group to estimate our minimum required sample size.
- questionnaires should be described in the introduction
- Thank you for this comment. All details related to the study questionnaires are described in the method section- under questionnaire tool heading, see page 3, lines 132-161.
Discussion:
- no comparison with the previous two studies - what is new, what is known and what differs.
- Thank you for this comment. Based on the reviewer comment, we have now enriched our discussion with more comparison to previous studies and interpretation (see pages 9-10, lines 276-303) and provided further recommendations for policymakers, healthcare professionals, government, and future research, see page11, lines 344- 376. Besides, we have now added in the introduction section justification for the current study and limitations of previous study to highlight what is known and what differs, see pages 2-3, lines 97-107.
- Authors didn't critically evaluate their work. All limitations are described in one line.
- Thank you for this comment. Based on the reviewer comment we have now added further study limitations to critically evaluate our work, see pages 11-12, lines 388-398.
- Some claims in the text are not suitable for a scientific manuscript.
E.g. line 265-266 "All of these issues can be resolved simply by raising awareness of epilepsy and including those who have it in educational sessions." Unfortunately, it is not as simple as the authors claim.
- Thank you for this comment. Based on the reviewer comment we have now rephrased this sentence and added more explanation, see page 11, lines 344-378.
Line 271-273, they call for action from the government. However, there is no data, and the authors did not provide any reference to support this.
- Thank you for this comment. Based on the reviewer comment we have now added more detailed recommendation from our study supported with relevant evidences, see page 11, lines 344-378.
Round 2
Reviewer 1 Report
The authors have answered well the comments. I think that the manuscript in present form is suitable for publication
Author Response
Thank you for confirming that no further changes are needed
Reviewer 3 Report
Authors improved the manuscript. However, the last few paragraphs in which they request substantial involvement of the government is not supported by the facts. It sounds more like an activism, not as a scientific paper. This should be improved or deleted.
Author Response
Thank you for your valuable time and effort in reviewing our paper. We aimed to provide some recommendations for the government to help in increasing the awareness of the general public about epilepsy. However, as we do not have references to support our recommendations, we followed the reviewer's recommendation and deleted the paragraph that request substantial involvement of the government, lines 378-384.
The remaining paragraphs provide general recommendations to policymakers and healthcare professionals.